# Next-Generation Sequencing of Connective Tissue Genes in Patients with Classical Ehlers-Danlos Syndrome

**Anna Junkiert-Czarnecka** [1,*] , **Maria Pilarska-Deltow** [1], **Aneta Bąk** [1], **Marta Heise** [1], **Anna Latos-Bieleńska** [2], **Jacek Zaremba** [3], **Alicja Bartoszewska-Kubiak** [1] and **Olga Haus** [1]

1   Department of Clinical Genetics, Faculty of Medicine, Collegium Medicum in Bydgoszcz, Nicolaus Copernicus University in Toruń, 85-094 Bydgoszcz, Poland; mpilarska@cm.umk.pl (M.P.-D.); aneta.bak@cm.umk.pl (A.B.); martaheise@cm.umk.pl (M.H.); abkubiak@cm.umk.pl (A.B.-K.); haus@cm.umk.pl (O.H.)
2   Department of Medical Genetics, Poznan University of Medical Sciences, 60-352 Poznan, Poland; alatos@ump.edu.pl
3   Department of Genetics, Institute of Psychiatry and Neurology, 02-957 Warsaw, Poland; zaremba@ipin.edu.pl
*   Correspondence: ajczarnecka@cm.umk.pl; Tel.: +48-52-585-3567

**Abstract:** Background: Ehlers-Danlos syndrome (EDS) is a common non-inflammatory, congenital connective tissue disorder. Classical type (cEDS) EDS is one of the more common forms, typically caused by mutations in the *COL5A1* and *COL5A2* genes, though causative mutations in the *COL1A1* gene have also been described. Material and methods: The study group included 59 patients of Polish origin, diagnosed with cEDS. The analysis was performed on genomic DNA (gDNA) with NGS technology, using an Illumina sequencer. Thirty-five genes related to connective tissue were investigated. The pathogenicity of the detected variants was assessed by VarSome. Results: The NGS of 35 genes revealed variants within the *COL5A1*, *COL5A2*, *COL1A1*, and *COL1A2* genes for 30 of the 59 patients investigated. Our panel detected no sequence variations for the remaining 29 patients. Discussion: Next-generation sequencing, with an appropriate multigene panel, showed great potential to assist in the diagnosis of EDS and other connective tissue disorders. Our data also show that not all causative genes giving rise to cEDS have been elucidated yet.

**Keywords:** Ehlers-Danlos syndrome; collagen; connective tissue; sequencing; NGS; Polish patients

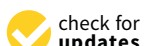



## 1. Introduction

Ehlers-Danlos syndrome (EDS) is a heterogeneous group of heritable connective tissue disorders. The 2017 International Classification of EDS recognizes 13 subtypes caused by pathogenic variants in 19 different genes, encoding different types of collagen or protein involved in collagen metabolism (Table 1). The most abundant types of EDS are classical (cEDS), vascular (vEDS), and hypermobile (hEDS); all EDS types, except hEDS, have their genetic backgrounds determined. According to the newest classification, classical EDS is inherited as an autosomal dominant disorder caused by mutations in *COL5A1*, *COL5A2*, or c.934C>T in *COL1A1*. For the clinical diagnosis of cEDS, major and minor criteria were established. cEDS should be suspected when skin hyperextensibility and atrophic scars are present (major criterion 1), together with joint hypermobility assessed with the Beighton score (major criterion 2), and/or with at least three minor criteria (easy bruising, soft, doughy skin, skin fragility, molluscoid pseudotumors, subcutaneous spheroids, hernias (or history thereof), epicanthal folds, complications of joint hypermobility (e.g., sprains, dislocations/subluxations, pain, pes planus), and family history of a first-degree relative who meets clinical criteria) [1,2]. Clinical features included in EDS classification are only one part of the disabilities recognized in EDS patients. In addition to major and minor phenotype criteria, patients' clinical picture also contains dysfunction of the gastrointestinal, cardiovascular, immune, neural, and other systems. The clinical symptoms may differ

among patients within the same family and may occur at different ages with various intensities. It is worth noting that not all cEDS patients with a pathogenic variant in the *COL5A1* or *COL5A2* gene fulfil the criteria for classical type EDS diagnosis according to the International Classification [2] (Table 1). Some may share their symptoms with other types of EDS (mainly with hypermobile, classical-like, or vascular) or other connective tissue disorders, and establishing their diagnosis is possible only after molecular investigation.

**Table 1.** The 2017 International Classification of Ehlers-Danlos Syndrome [2].

| N. | EDS Type | Genetic Basis | Protein |
|---|---|---|---|
| 1. | Classical EDS (cEDS) | *COL5A1, COL5A2, COL1A1* (c.934C>T) | Type V collagen Type I collagen |
| 2. | Classical-like EDS (clEDS) | *TNXB* | Tenascin XB |
| 3. | Cardiovalvular EDS (cvEDS) | *COL1A2* | Type I collagen |
| 4. | Vascular EDS (vEDS) | *COL3A1, COL1A1* (c.934C>T, c.1720C>T, c.3227C>T) | Type III collagen, Type I collagen |
| 5. | Hypermobile (hEDS) | Unknown | Unknown |
| 6. | Arthrochalasia (aEDS) | *COL1A1, COL1A2* | Type I collagen |
| 7. | Dermatosparaxis (dEDS) | *ADAMTS2* | ADAMTS-2 |
| 8. | Kyphoscoliotic EDS (kEDS) | *PLOD1, FKBP14* | LH1, FKBP14 |
| 9. | Brittle cornea syndrome (BCS) | *ZNF469, PRDM5* | ZNF469, PRDM5 |
| 10. | Spondylodysplastic (sEDS) | *B4GALT7, B3GALT6, SLC39A13* | b4GalT7, b3GalT6, SLC39A13 |
| 11. | Musculo-contractural EDS (mcEDS) | *CHST14, DSE* | CHST14, DSE |
| 12. | Myopathic EDS (mEDS) | *COL12A1* | Type XII collagen |
| 13. | Periodontal (pEDS) | *C1R, C1S* | C1r, C1s |

Diagnosis of cEDS must be established based on clinical criteria and confirmed by molecular analysis. Genetic testing was primarily based on single-gene testing by Sanger sequencing (*COL5A1*, *COL5A2*, and c.934C>T in *COL1A1*). However, nowadays, because of phenotypic heterogeneity and clinical overlapping among EDS types and between EDS and other connective tissue disorders, single-gene testing conducted after clinical evaluation is often not definitive and leaves EDS patients without a molecular diagnosis. Molecular testing with next-generation sequencing (NGS) and a multigene panel, containing EDS-related and other connective tissue-associated genes (for diagnosis specifying), seems to be a very adequate method.

In the present study, we evaluated the molecular background of Polish cEDS patients by NGS with an Illumina connective tissue gene panel.

## 2. Materials and Methods

The study group included 59 patients of Polish origin, women and men, aged 3–63 years (median, 23). Patients were enrolled in the investigation by experienced clinical geneticists according to the 2017 International Classification of the Ehlers-Danlos Syndrome diagnostics criteria. Joint hypermobility was evaluated according to the Beighton scale.

All cEDS patients or their parents provided informed consent. The study was also approved by the Ethics Committee of the Collegium Medicum in Bydgoszcz, Nicolaus Copernicus University in Torun, Poland.

The analysis was performed on genomic DNA (100 ng) (gDNA) extracted from leukocytes with a QIAamp DNA Mini Kit (Qiagen, Hilden, Germany) using standard procedures. The molecular investigation was made using NGS technology by Illumina

(NextSeq 550) with customer connective tissue panel genes (NimbleGen, Roche): *COL5A1, COL5A2, COL3A1, COL1A1, COL1A2, TNXB, ADAMTS2, PLOD1, FKBP14, ZNF469, PRDM5, B4GALT7, B3GALT6, SLC39A13, CHST14, DSE, COL12A1, C1R, C1S, COL6A1, COL6A2, COL6A3, COL9A1, COL9A2, FBN1, FBN2, FLNA, FLNB, ELN, NOTCH1, MYH11, MYLK, TGFB2, TGFB3, TGFBR1*. The algorithm used for alignment and variant calling was Burrows-Wheeler Aligner (BWA), variant identification was made by Genome Analysis Toolkit (GATK). The pathogenicity of detected variants was assessed by Varsome (10.0) [3]. All pathogenic and likely pathogenic variants were confirmed by Sanger sequencing (ABI3130XL with BigDye and XTerminator (ThermoFisher, Waltham, MA, USA).

## 3. Results

Among 59 patients (from 56 families), molecular changes in the *COL5A1* gene were found in 20. One patient was a carrier of the *COL5A2* gene variant. Furthermore, three patients were carriers of alterations in *COL1A1*, and six patients were carriers of *COL1A2* gene variants. In 29 patients, no variants in tested genes were found.

### 3.1. Variants in COL5A1 and COL5A2 Genes

Seven patients were carriers of pathogenic or likely pathogenic *COL5A1* alterations, twelve were carriers of benign or likely benign variants, and one of them was a carrier of the VUS variant. In one patient, a likely pathogenic alteration in *COL5A2* was detected. Variants were described in Table 2. Clinical symptoms characteristics of all patients are described in Table S1 (Supplementary Data).

In Patient 1 (11-year-old girl), a splice site alteration (c.1989+1G>T) was identified. This variant was described previously by Mitchell et al.; however, in their patient, guanine was substituted by adenine at position c.1989 (c.1989+1G>A) [4]. In our patient, guanine was replaced by thymine (c.1989+1G>T). In both cases, substitution took place at the site critical for the splicing process. Therefore, according to Varsome, c.1989+1G>T may also be pathogenic.

Variant c.1273_1276dupAGTC (p.Ser426Ter), detected in a 6-year-old girl (Patient 2), was not described up to now in cEDS patients, and according to Varsome analysis, it is likely pathogenic.

In Patients 3, 4, and 5 a frameshift variant c.5021_5021delC (p.Thr1674Lysfs*55) was detected. This alteration had not yet been described in cEDS patients. Bioinformatic analysis testified its pathogenicity. Patient 3 was the mother of Patients 4 and 5.

A frameshift variant c.4050dupC (p.Gly1351Argfs*814) in Patient 6 (mother) and Patient 7 (daughter) was detected. This is a known pathogenic variant described by Symoens et al. [5].

**Table 2.** Variants in *COL5A1* and *COL5A2* genes detected in the investigated group.

| Patient | Gene | Variant | Protein | dbSNP | Varsome |
|---|---|---|---|---|---|
| 1. | *COL5A1* | c.1989+1G>T | splice variant | not reported | Pathogenic |
| 2. | *COL5A1* | c.1273_1276dupAGTC | p.Ser426Ter | not reported | Likely Pathogenic |
| 3. | *COL5A1* | c.5021delC | p.T1674Kfs*55 | not reported | Pathogenic |
| 4. | *COL5A1* | c.5021delC | p.T1674Kfs*55 | not reported | Pathogenic |
| 5. | *COL5A1* | c.5021delC | p.T1674Kfs*55 | not reported | Pathogenic |
| 6. | *COL5A1* | c.4050dupC | p.Gly1351Argfs*814 | not reported | Likely Pathogenic |
| 7. | *COL5A1* | c.4050dupC | p.Gly1351Argfs*814 | not reported | Likely Pathogenic |
| 8. | *COL5A1* | c.1726C>T | p.Pro576Ser | rs763246328 | Likely Benign |
|  | *COL6A3* | c.6930+5G>A | splice variant | rs749037028 | VUS |
| 9. | *COL5A1* | c.944C>T [#] | p.Thr315Met | rs145093766 | Likely Benign |

**Table 2.** *Cont.*

| Patient | Gene | Variant | Protein | dbSNP | Varsome |
|---|---|---|---|---|---|
| 10. | *COL5A1* | c.3023C>T | p.Thr1008Met | rs199735010 | Likely Benign |
| 11. | *COL5A1* | c.3023C>T | p.Thr1008Met | rs199735010 | Likely Benign |
| 12. | *COL5A1* | c.3023C>T | p.Thr1008Met | rs199735010 | Likely Benign |
| 13. | *COL5A1* | c.3023C>T | p.Thr1008Met | rs199735010 | Likely Benign |
| 14. | *COL5A1* | c.3398G>A | p.Arg1133Gln | rs759580799 | Likely Benign |
| 15. | *COL5A1* | c.1089C>G [#] | p.Asn363Lys | rs773870913 | Likely Benign |
| 16. | *COL5A1* | c.193C>T [#] | p.Arg65Trp | rs139468527 | Benign |
| | *COL5A1* | c.514G>T [#] | p.Val172Phe | rs150147262 | Likely Benign |
| 17. | *COL5A1* | c.367C>G [#] | p.Gln123Glu | rs142114921 | Likely Benign |
| 18. | *COL5A1* | c.4483G>A [#] | p.Gly1495Ser | not reported | VUS |
| 19. | *COL5A1* | c.2588A>T [#] | p.Glu863Val | rs139788610 | Benign |
| | *COL5A1* | c.3418G>A [#] | p.Val1140Met | rs149616140 | Benign |
| 20. | *COL5A1* | c.3418G>A [#] | p.Val1140Met | rs149616140 | Benign |
| 21. | *COL5A2* | c.2555G>A | p.Gly852Asp | not reported | Likely Pathogenic |

Legend: [#] variants described previously in [6].

In Patient 8, a 56-year-old woman, variant c.1726C>T (p.Pro576Ser) in *COL5A1*, likely benign, and a second VUS splice site variant c.6930+5G>A in *COL6A3* were found.

Other variants detected in *COL5A1* were benign or likely benign. The clinical description of those patients was included in Table S1 (Supplementary Data).

In Patient 21, a 23-year-old woman, a likely pathogenic variant c.2555G>A (p.Gly852Asp) in *COL5A2* was detected. This variant was not recorded in the LOVD database and was not, up to now, detected in cEDS patients.

### 3.2. Variants in COL1A1 and COL1A2 Genes

Molecular changes in *COL1A1* and *COL1A2* genes were detected in nine patients. Three of them were carriers of variants in *COL1A1* and six variants in the *COL1A2* gene. Variants information was described in Table 3. Clinical symptoms of all patients are described in Table S1 (Supplementary Data).

**Table 3.** Variants in *COL1A1* and *COL1A2* genes detected in the investigated group of patients with cEDS clinical features.

| Patient | Gene | Variant | Protein | dbSNP | Varsome |
|---|---|---|---|---|---|
| 22 | *COL1A1* | c.2451T>C | p.Pro817= | rs374465457 | Likely Pathogenic |
| 23 | *COL1A1* | c.517G>A | p.Gly173Arg | rs193922157 | VUS |
| 24 | *COL1A1* | c.1984-5C>A | splice variant | rs66592376 | Benign |
| 25 | *COL1A2* | c.601C>A | p.Pro201Thr | not reported | VUS |
| | *COL1A2* | c.661G>A | p.Gly221Ser | not reported | VUS |
| 26 | *COL1A2* | c.3706A>G | p.Ser1236Gly | rs781184808 | VUS |
| 27 | *COL1A2* | c.2776C>T | p.Arg926Cys | rs745363291 | VUS |
| 28 | *COL1A2* | c.118C>A | p.Pro40Thr | rs1363689462 | VUS |
| 29 | *COL1A2* | c.2642A>C | p.Glu881Ala | rs751201659 | VUS |
| 30 * | *COL1A2* | c.3313G>A | p.Gly1105Ser | rs139851311 | VUS |

Legend: * in Patient 30, two aditional variants were detected: NOTCH1 (c.3142C>T, p.Pro1048Ser, rs770521856, VUS) and COL6A3 (c.4184G>A, p.Arg1395Gln, rs80272723, Benign).

Patient 22 was an 18-year-old woman with a variant c.2451T>C (p.Pro817=) in *COL1A1*. This silent variant was assessed as a likely pathogenic by Varsome. It was not detected, up to now, in the LOVD database and was not described in Ehlers-Danlos Syndrome or Osteogenesis Imperfecta patients (OI). However, silent mutations in *COL1A1* were already detected in Czech and Japanese OI patients [7,8]. The father of Patient 22, who presented clinical symptoms similar to his daughter but in a milder form was also a carrier of c.2451T>C.

Patient 23 was a 20-year-old woman with a variant c.517G>A (p.Gly173Arg) in *COL1A1.* Varsome determined this variant as VUS, but in LOVD, it was described as potentially pathogenic in a patient with suspected hypophosphatasia (patient with clinical symptoms but without pathogenic variants in *ALPL* gene) [9]. The patient did not report bone fractures or any other bone abnormalities typical for hypophosphatasia. The patient's mother and brother presented similar symptoms, but they did not have molecular testing for the c.517G>A variant.

In Patient 24, variant c.1984-5C>A was found, described in LOVD as not pathogenic.

Patient 30 was a 14-year-old boy with the c.3313G>A variant in *COL1A2* (VUS), c.3142C>T in *NOTCH1* (VUS), and c.4184G>A in *COL6A3* (benign). The patient also has neurofibromatosis type 1 (von Recklinghausen's disease) with café au lait spots.

In patients, 25–29 VUS variants in *COL1A2* were found. Up to now, these variants have not been described in LOVD, and neither were found in EDS or OI patients.

## 4. Discussion

Ehlers-Danlos syndrome is a heritable, non-inflammatory congenital disorder of connective tissue. Clinical diagnosis of EDS must be made according to directives included in The 2017 EDS International Classification based on major and minor criteria. Minimal requirements for diagnosing classical EDS (cEDS) are skin hyperextensibility and atrophic scarring plus either generalized joint hypermobility or three out of nine minor criteria. According to this classification, clinical diagnosis should be confirmed by molecular testing of *COL5A1*, *COL5A2*, and c.934C>T *COL1A1* alterations [1,2]. Except for the clinical symptoms included in the classification, one of the features that most often occur in cEDS patients is their clinical diversity and overlapping of the clinical picture with other connective tissue disorders. This study considered ours and other researchers' experiences that cEDS patients with pathogenic variants in *COL5A1* or *COL5A2* do not always have a phenotype compatible with criteria in the 2017 EDS International Classification; thus, not all of the patients tested here fulfilled the cEDS clinical criteria [10,11].

Among 59 investigated patients, 20 variants in *COL5A1* were found. In 7 of them, there were pathogenic or likely pathogenic nucleotide changes; in the remaining 13, they were benign or likely benign. In one patient, a pathogenic variant in the *COL5A2* gene was found. We haven't seen any clinical differences between the mutation carriers and not the carrier. We also haven't observed any impact of genetic factors on the development of the clinical symptoms.

The next group included patients with *COL1A1* and *COL1A2* gene variants, which, in addition to *COL5A1* and *COL5A2*, are well-known causes of cEDS or cEDS overlapping with OI [12–14]. We found three patients with likely pathogenic, VUS, and benign variants in *COL1A1* and six patients with VUS variants in *COL1A2*. It is worth underlining that one VUS variant in *COL1A1* and three VUS variants in *COL1A2* were substitutions of glycine amino acid, which plays an essential role in the stability of the collagen molecule. Similarly, in *COL5A1* (c.1273_1276dupAGTC, c.5021delC) and *COL1A1* (c.2451T>C), new variants, not described in EDS patients were detected. To determine their role in EDS development, functional analyses are needed, especially for silent variants in *COL1A1.*

Next-generation sequencing with a multigene panel was applied in this investigation and is a method with great potential in EDS or other connective tissue disorder diagnostics. Nevertheless, we have considered that it is not an absolute method. Using NGS for genomic investigation, we cannot perform some tests, e.g., the null allele test for investigating the

well-known mechanism of EDS development [15]. Other genetic changes in EDS patients not covered in this investigation are deletions or insertions, which can be detected by MLPA or aCGH methods. In Ritelli et al., an investigation among 40 patients, one of the duplications of exons 1–11 in *COL5A1* was detected (probe for *COL5A2* were not available) [16]. In the investigation of Kuroda et al., cEDS patients with alterations in *COL5A1* (deletion of exons 2–11 and duplication of exons 12–65) were determined identified by aCGH [17]. All patients analyzed in the present study need wider diagnostics because we cannot exclude DNA changes other than those detected by NGS. We also have to consider that some of our patients may have hypermobile-type EDS, and genetic background of this type is not determined. Hypermobile patients with a phenotype similar to classical EDS were described by Castori et al. [18]. Authors presented hEDS patients with mucocutaneous abnormality similar to the skin abnormalities described in cEDS classification (atrophic scars, hyperextensible, soft, and velvety skin). The similarity in cEDS and hEDS phenotype in some patients may lead to inaccurate patient classifications. This fact may also clarify the reason for the lower percentage of *COL5A1/COL5A2* mutations in our group compared to those assessed by Symoens et al. [5] and Ritelli [16], where the percentages of deleterious variants were determined as 90 and 93, respectively.

## 5. Conclusions

Years of investigations showed that Ehlers-Danlos syndrome is a disorder with a very complex phenotype and highly complicated genotype.

**Supplementary Materials:** The following supporting information can be downloaded at: https://www.mdpi.com/article/10.3390/cimb44040099/s1, Table S1: title Clinical findings of the 59 patients with cEDS.

**Author Contributions:** Conception and the preparation of the manuscript, A.J.-C. and O.H.; molecular investigation, A.J.-C., M.P.-D., A.B., M.H. and A.B.-K.; patient enrolment, O.H., A.L.-B. and J.Z. All authors have read and agreed to the published version of the manuscript.

**Funding:** This investigation was supported by the Nicolaus Copernicus University Statutory Research Grant number 286.

**Institutional Review Board Statement:** The study was approved by the Ethics Committee of the Collegium Medicum in Bydgoszcz, Nicolaus Copernicus University in Torun, Poland.

**Informed Consent Statement:** Informed consent was obtained from all subjects involved in the study.

**Data Availability Statement:** All data and materials not included in the manuscript are available upon request. Gene variants are available in LOVD.

**Acknowledgments:** The authors would like to thank all patients and their families for participating in the investigation.

**Conflicts of Interest:** The authors declare no conflict of interest.

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
