# Peer review of "Next-Generation Sequencing of Connective Tissue Genes in Patients with Classical Ehlers-Danlos Syndrome"

_cimb, doi:10.3390/cimb44040099_

Round 1

Reviewer 1 Report

It is a very interesting paper and some variants discovered were related to the clinical features of the patients. I understand the difficulties in writing a manuscript in a non-native language, however the interest in the paper is lost due to poor sentence structure and misuse of adjectives. The exact genes that were sequenced should be published and not referred to "genes such as". For example, I have taken the time to improve the abstract with proper English:

Abstract:

Background: Ehlers-Danlos syndrome (EDS) is a common non-inflammatory, congenital connective tissue disorder. Classical type (cEDS) is one of the more common forms of EDS, typically due to mutations in COL5A1 and COL5A2 genes though causative mutations in the COL1A1 gene have also been described.

Materials and methods: The study group included 59 patients of Polish origin who have been diagnosed with cEDS. The analysis was performed on genomic DNA (gDNA) with the NGS technology by Illumina. ?? genes related to connective tissue were investigated. The pathogenicity of detected variants was assessed by VarSome.

Results: NGS of ?? genes revealed variants within the COL5A1, COL5A2, COL1A1 COL1A2 genes for 30 of the 59 patients investigated. No sequence variations were detected by our panel for the remaining 29 patients.

Discussion: Next Generation Sequencing with an appropriate multigene panel shows great potential to assist in the diagnosis of EDS and other connective tissue disorders. Our data also shows that all causative genes giving rise to cEDS have yet to be been elucidated. 

This is just an example. Editing for clarity in English is strongly recommended before being accepted for publication. 

I am Polish/Canadian and my comments have no bias nor prejudice.

Author Response

Dear Reviewer 1,

Thank you very much for your feedback and your help. I’ve addressed your change recommendations as follows.

Best Regards,

Anna Junkiert-Czarnecka

Reviewer 2 Report

The authors summarize genetic findings for a cohort of 59 patients of Polish origin with classic Ehlers-Danlos syndrome. The analysis was performed with a multigene panel and next-generation sequencing. The study contributes several variants not previously reported to be associated with cEDS to the literature and the LOVD database. Overall, the manuscript is well written and sound, but it could benefit from additional details, clarification and organization.

Specific comments:

The conclusion is unclear. Are you saying that the genetic causes of cEDS are heterogenous? Especially in light of the variants identified in this study that have not been previously associated with cEDS?

The methods section needs additional details provided. How much DNA was used for library prep? What Illumina instrument and flow cell was used? Were all samples batched on a single run? Bioinformatics details are necessary – what algorithms used for alignment and variant calling? What regions were analyzed? Were raw VCF files uploaded to Varsome for interpretation or were they first annotated and then manually classified via Varsome? What version of Varsome was used? At least one variant, COL1A2(NM_000089.4):c.3313G>A, has a different classification with the current version (Version: 11.1.10) compared to what is listed in the table. Please specify transcripts variants were annotated against; e.g. COL1A2(NM_000089.4).  Details necessary for Sanger – reagents, instruments, software etc. Did any NGS variants fail to validate by Sanger?

You mention deletions/duplications detected in other studies, did you attempt del/dup calling on your NGS data? Some callers perform well for NGS panel data - https://www.nature.com/articles/s41431-020-0675-z

The results section is a little confusing as you describe benign variants, but these are unlikely to account for the patients phenotype. Is it unusual to find pathogenic variants in only 8 of 59 patients? Were there variants in the other genes on the panel for the patients without described variants? Were other benign variants in these genes identified and not discussed? If so, is there a reason to focus on these benign variants? The identification of several variants not previously reported to be associated with cEDS seems like it could be highlighted more; are the phenotypes of these patients unique etc.

Table numbering seems off. No table 3 and supplemental not in text.

Line 72 – “…by Illumina with connective tissue panel genes such as…”

I am unable to locate this panel on Illumina’s site. Can you specify which panel was used (e.g. TruSight, AmpliSeq for Illumina, etc.). Also state if these are the only genes on the panel or are there additional genes that weren’t analyzed. The regions of interest targeted and analyzed could be provided as a supplemental file; are there regions of the genes not targeted, intronic boundaries etc.

Line 87 – “Clinical symptoms of all patients were described in table.”

No table number listed. supplemental table 1?

Line 106 – “Other variants detected in COL5A1 were benign or likely benign. Clinical description of those patients was included in table 4.”

This should be supplemental table 1?

Line 131 – “…in LOVD as rather not pathogenic”

This variant has a range of classifications in LOVD; from benign to likely pathogenic

https://databases.lovd.nl/shared/view/COL1A1?search_VariantOnGenome%2FDBID=%3D%22COL1A1_000423%22

Author Response

Dear Reviewer 2,

Thank you very much for your feedback and your help. I’ve addressed your change recommendations as follows.

Best Regards,

Anna Junkiert-Czarnecka
